# Liquid Biopsy in B and T Cell Lymphomas: From Bench to Bedside

**DOI:** 10.3390/ijms26104869

**Published:** 2025-05-19

**Authors:** Mohammad Almasri, Nawar Maher, Bashar Al Deeban, Ndeye Marie Diop, Riccardo Moia, Gianluca Gaidano

**Affiliations:** 1Division of Hematology, Department of Translational Medicine, Università del Piemonte Orientale and Azienda Ospedaliero-Universitaria Maggiore della Carità, 28100 Novara, Italy; mohammad.almasri@uniupo.it (M.A.); nawar.maher@uniupo.it (N.M.); bashar.aldeeban@uniupo.it (B.A.D.); marie.diop@uniupo.it (N.M.D.); riccardo.moia@uniupo.it (R.M.); 2Division of Hematology, Department of Translational Medicine, Università del Piemonte Orientale and Azienda Ospedaliero-Universitaria di Alessandria, 56121 Alessandria, Italy

**Keywords:** liquid biopsy, cell-free DNA, circulating tumor DNA, B-cell lymphoma, T-cell lymphoma

## Abstract

Liquid biopsy through the analysis of circulating tumor DNA (ctDNA) is emerging as a powerful and non-invasive tool complementing tissue biopsy in lymphoma management. Whilst tissue biopsy remains the diagnostic gold standard, it fails to detect the molecular heterogeneity of the tumor’s multiple compartments and poses challenges for sequential disease monitoring. In diffuse large-B-cell lymphoma (DLBCL), ctDNA facilitates non-invasive genotyping by identifying hallmark mutations (e.g., *MYD88*, *CD79B*, *EZH2*), enabling molecular cluster classification. Dynamic changes in ctDNA levels during DLBCL treatment correlate strongly with progression-free survival and overall survival, underscoring its value as a predictive and prognostic biomarker. In Hodgkin’s lymphoma, characterized by a scarcity of malignant cells in tissue biopsies, ctDNA provides reliable molecular insights into tumor biology, response to therapy, and relapse risk. In primary central nervous system lymphoma, the detection of *MYD88* L265P in ctDNA offers a highly sensitive, specific, and minimally invasive diagnostic option. Likewise, in aggressive T-cell lymphomas, ctDNA supports molecular profiling, aligns with tumor burden, and shows high concordance with tissue-based results. Ongoing and future clinical trials will be critical for validating and standardizing ctDNA applications, ultimately integrating liquid biopsy into routine clinical practice and enabling more personalized and dynamic lymphoma care.

## 1. Introduction

Lymphomas represent a heterogeneous group of malignancies that arise from the clonal proliferation of B-cells, T-cells, and natural killer cells at different stages of maturation. These hematological malignancies demonstrate marked differences in disease biology, prognosis, and response to treatment [1]. Advancements in precision medicine have revolutionized the field of hematoncology, enabling personalized treatment strategies tailored to the molecular profile for a sizeable fraction of patients with a blood cancer [2].

In many lymphomas, understanding the genetic and molecular characteristics of the tumor is critical for accurate diagnosis, risk stratification, and therapeutic decision-making [3,4]. Conventionally, this information is obtained through a tissue biopsy, which, though being essential for diagnosis, has limitations [5]. Tissue biopsy is an invasive procedure that presents challenges for obtaining serial samples, which are necessary for real-time disease monitoring and therapeutic assessment. Moreover, because tissue biopsy typically samples a single tumor site, it may provide incomplete genomic information and fails to capture the complexity of tumor heterogeneity across different anatomical compartments [5,6]. Insights from solid tumors, where liquid biopsy is increasingly used to address similar limitations, provide valuable translational context [7]. In these settings, ctDNA has been effectively utilized to monitor treatment resistance and disease progression, despite challenges in assay sensitivity and standardization [8]. These parallels emphasize the potential of ctDNA to revolutionize lymphoma management and underscore the need to overcome analytical and clinical hurdles to enable its broader adoption in hematologic oncology.

In recent years, liquid biopsy has emerged as a possible complementary tool integrating tissue biopsy for lymphoma molecular diagnostics and management. This minimally invasive approach enables the identification of disease-related biomarkers through accessible procedures such as blood drawing or the collection of body fluids like saliva, urine, cerebrospinal fluid (CSF), and stool. Liquid biopsy offers several advantages over conventional tissue biopsy, including the ability to capture tumor heterogeneity at multiple sites, support real-time disease monitoring, and allow for longitudinal sampling across treatment [9,10,11].

Liquid biopsy includes several tumor-derived components circulating in body fluids that provide important information about tumor biology. Initially, research in this field focused on circulating tumor cells (CTCs), but it has been expanded to other biomarkers including circulating tumor DNA (ctDNA), cell-free RNA (cfRNA), extracellular vesicles (EVs), and tumor-educated platelets (TEPs) (Figure 1) [12,13]. Among these biomarkers, ctDNA has been the most extensively studied in hematological malignancies as a tool for disease genotyping and dynamic monitoring of the disease response during therapy [14]. In this review, we will explore the current applications of ctDNA in B-cell and T-cell lymphomas, highlighting its potential role in enabling a more personalized treatment approach for each patient.

### 1.1. Biological Basis of ctDNA

Cell-free DNA (cfDNA) consists of fragmented DNA molecules that circulate in the bloodstream, originating from different biological processes, such as apoptosis, necrosis, or active release from living cells [15,16]. cfDNA was first identified in human blood in 1948, and fetal-derived cfDNA in maternal plasma was observed in 1977 [17]. Studies have shown that cancer patients generally have higher levels of cfDNA compared to healthy individuals due to increased DNA shedding from tumor cells into the circulation [14].

cfDNA fragmentation follows a structured pattern rather than a random occurrence [18]. In healthy individuals, most cfDNA arises from apoptotic cells and has a typical fragment size of around 160–170 base pair (bp), reflecting how DNA is packaged around nucleosomes. In contrast, tumor-derived cfDNA, known as ctDNA, often consists of shorter fragments, averaging around 143 bp [19]. The biological reasons behind these fragmentation differences remain unclear [20].

While cfDNA can originate from both normal and malignant cells, ctDNA is specifically released from tumor-related sources. ctDNA is highly dynamic and rapidly cleared from the bloodstream, with an estimated half-life of 1 to 2 h, thus providing a real-time snapshot of the tumor molecular profile [21,22]. The levels of ctDNA in the bloodstream vary widely among patients and are influenced by factors such as cancer type, tumor burden, disease progression, and treatment response [23]. In addition to its levels, ctDNA carries tumor-specific genetic and epigenetic alterations, including mutations, chromosomal rearrangements, and DNA methylation patterns [24]. These unique molecular characteristics render ctDNA a valuable tool for cancer diagnosis, monitoring treatment response, detecting minimal residual disease (MRD), and guiding personalized therapeutic strategies [25].

### 1.2. Detection and Analytical Techniques

The effectiveness of ctDNA analysis largely depends on the type of biological sample utilized. Theoretically, any body fluid containing cfDNA could be a potential source for liquid biopsy, including saliva, urine, cerebrospinal fluid, and stool. However peripheral blood (PB) remains the most preferred and widely used source of ctDNA due to its accessibility and convenience in collection (Figure 1) [5,26].

While serum can be used for ctDNA extraction, plasma is generally preferred for ctDNA analysis. Serum samples carry a higher risk of contamination from genomic DNA released by lysed blood cells during the coagulation process in the tube. Plasma reduces this risk, providing a higher-purity sample for ctDNA isolation [27,28]. Blood samples should ideally be processed within six hours of collection using EDTA tubes to avoid contamination from leukocytes and other cellular components. Specialized cell preservation tubes, such as Streck BCT^TM^ or CellSave^TM^ tubes, have been developed to stabilize ctDNA and prevent cell rupture for up to 14 days, which is particularly useful when transporting samples to distant laboratories (Figure 1) [29,30].

A critical step in plasma preparation is the use of double-spin centrifugation, which effectively separates plasma from leukocytes and ensures a high-quality plasma sample, improving ctDNA isolation [31,32]. However, variations in storage conditions, sample processing methods, and time elapsed before analysis can all influence the overall quality and quantity of ctDNA. Careful standardization of these factors is essential for obtaining reliable and consistent results [33]. The accuracy and sensitivity of ctDNA detection techniques depends not only on the quality of the sample but also on the technologies used for ctDNA extraction and analysis. Since ctDNA constitutes only a small fraction of cfDNA, obtaining an adequate amount of ctDNA can be challenging, particularly when tumor burden is low. Consequently, larger blood volumes, typically around 10 mL, are often required to obtain enough ctDNA for analysis [34,35].

Once ctDNA is isolated, the detection process is carried out using highly sensitive analytical techniques, such as digital PCR (dPCR), droplet digital PCR (ddPCR), and next-generation sequencing (NGS) (Figure 1). dPCR and ddPCR are both powerful techniques known for their high specificity and ability to detect rare mutations at low allele frequencies [36]. These methods are particularly useful for monitoring known mutations, which makes them particularly valuable in monitoring tumor progression or detecting specific biomarkers in established cancers [20]. However, their main limitation lies in their restricted ability to identify a small set of predefined mutations, which limits their broader applicability. By contrast, NGS provides a comprehensive analysis of multiple genetic variations simultaneously, making it ideal for discovering new mutations, structural variations, and other genetic changes that might not be detected by PCR-based methods. NGS can identify a broad range of genetic alterations, including point mutations, insertions, and deletions, copy number variations (CNVs), and gene fusions, providing a more comprehensive picture of the tumor genetic landscape [37,38,39]. However, NGS requires higher sequencing depths to accurately detect low-frequency mutations. This can make NGS more costly and computationally intensive compared to PCR-based methods. Furthermore, the complexity of the data generated by NGS necessitates sophisticated bioinformatic tools to interpret the results correctly [40].

Recent innovations in sequencing technologies, including the implementation of unique molecular identifiers (UMIs) and error-correction algorithms, have improved NGS sensitivity [41]. These improvements enable more reliable detection of ctDNA at low concentrations. Additionally, emerging techniques, such as nanomaterial-based assays and artificial intelligence, are expected to further enhance ctDNA detection by improving sensitivity, reducing costs, and providing deeper understanding of the tumor heterogeneity, making ctDNA-based liquid biopsies an even more powerful tool for cancer diagnosis, monitoring, and personalized therapeutic strategies.

## 2. ctDNA in B-Cell Lymphoma

### 2.1. Diffuse Large-B-Cell Lymphoma

Diffuse large-B-cell lymphoma (DLBCL) is the most common subtype of non-Hodgkin’s lymphoma (NHL), representing approximately 30–35% of all cases. DLBCL displays significant biological heterogeneity, both across patients and across distinct tumor sites within the same individual [42,43]. The standard treatment for DLBCL traditionally includes the combination of rituximab with cyclophosphamide, doxorubicin, vincristine, and prednisone (R-CHOP), which leads to durable complete remissions (CRs) in over 60% of patients [44,45]. More recently, the combination of polatuzumab vedotin with rituximab, cyclophosphamide, doxorubicin, and prednisone (Pola-R-CHP) has also emerged as a frontline standard, particularly in patients with intermediate- to high-risk DLBCL based on the international prognostic index (IPI) score [44]. However, the heterogeneity in the molecular landscape of DLBCL presents challenges in assessing response to treatment, prognosis, and identification of patients with early relapsing/refractory disease after first-line chemoimmunotherapy [46].

Traditional diagnostic methods, such as tissue biopsies, often fail to fully capture this complexity, underscoring the need to explore new approaches for personalizing treatment and improving patient outcomes [6]. One promising approach for enhancing DLBCL management is the use of ctDNA, a powerful tool for the detection of genetic alterations that otherwise might be missed in the tissue biopsy. Advanced sequencing technologies, such as cancer personalized profiling by deep sequencing (CAPP-seq), have demonstrated that ctDNA can detect hallmark genetic alterations associated with DLBCL subtypes, including mutations of *MYD88, CD79B*, and *EZH2*, making it a useful tool for disease genotyping, enabling molecular classification of DLBCL and guiding targeted therapies. Studies have shown a high concordance between ctDNA genotyping and tissue biopsy for detecting genetic alterations in DLBCL (Figure 2) [6,47,48].

The molecular classification of DLBCL was established mainly on the tissue biopsy, enabling the identification of clinically relevant genetic subtypes with distinct features [49,50]. Wright et al. have developed the LymphGen algorithm, a probabilistic tool designed to assign DLBCL tumors to molecular clusters based on individual genomic profiles [50]. These clusters include MCD, BN2, N1, A53, and EZB and are defined by the enrichment of specific genetic aberrations and differ in their gene expression signatures and response to chemoimmunotherapy [50,51] (Table 1).

Recent evidence shows that LymphGen can also be applied reliably and effectively on ctDNA. A recent prospective study of 166 newly diagnosed DLBCL patients by Moia et al. has reported a 95.8% concordance rate in molecular cluster assignment between ctDNA and tissue biopsy, highlighting the reproducibility of molecular clustering on plasma [46]. As with tissue biopsy studies, patients assigned to the A53 and MCD clusters exhibited poorer outcomes, while those in the ST2 and BN2 clusters showed significantly better outcomes [46]. This suggests that ctDNA can serve as a reliable alternative for molecular characterization, particularly in cases for which the tissue biopsy is insufficient for the analysis.

Pretreatment ctDNA levels have been identified as a key prognostic biomarker in DLBCL. Several studies have linked higher baseline ctDNA levels to poorer progression-free survival (PFS), event-free survival (EFS), and overall survival (OS) [62,63,64]. In a large study by Kurtz et al. using CAPP-seq, baseline ctDNA concentration was found to be a stronger predictor of EFS than conventional prognostic factors, such as the cell of origin (COO), IPI score, and total metabolic tumor volume (TMTV) [52]. Moreover, ctDNA levels could also be integrated with molecular clusters identified on the liquid biopsy to further refine outcome prediction [46]. In fact, DLBCL patients assigned to the ST2 and BN2 molecular clusters are associated with favorable prognosis even in the presence of elevated ctDNA levels. This combined approach of integrating multiple ctDNA-derived biomarkers enhances the prognostic accuracy and allows for more precise risk stratification of DLBCL patients [46].

In addition to pretreatment prognostic value, the dynamic changes in ctDNA levels during therapy have been shown to reflect treatment response in DLBCL. Early molecular response (EMR), defined as a 2-log reduction in ctDNA levels after one cycle of therapy, and major molecular response (MMR), characterized by a 2.5-log reduction after two cycles, have been significantly correlated with better PFS and OS [65,66]. Kurtz et al. have documented that patients achieving EMR and MMR have significantly better clinical outcomes compared to patients with inferior ctDNA reduction. Moreover, ctDNA reduction remained a strong independent predictor of survival even after adjusting for standard prognostic factors, including IPI and COO [67]. These molecular response biomarkers offer an additional tool for monitoring treatment response and may integrate traditional methods such as PET/CT scans [68].

Beyond standard therapy, the utility of ctDNA extends to novel therapeutic approaches, such as chimeric antigen receptor (CAR) T-cell therapy. Zou et al. have documented that higher pre-infusion ctDNA levels and shorter ctDNA fragment sizes (<170 bp) correlate with inferior outcomes following CAR T-cell therapy. Additionally, ctDNA clearance at day 14 and day 28 post-infusion was associated with higher rates of 3-month CR and prolonged PFS and OS [69]. These findings suggest that ctDNA monitoring may serve as an early indicator of the efficacy of CAR T-cell therapy much earlier than conventional imaging scans and may aid in optimizing patient management. Moreover, integrated ctDNA profiling in CAR-T-cell-treated patients affected by relapsed/refractory (R/R) large-B-cell lymphoma (LBCL) has uncovered key resistance mechanisms, including molecular alterations of genes affecting B-cell identity (e.g., *PAX5*, *IRF8*), immune evasion *(CD274)*, and the tumor microenvironment *(TMEM30A)* [70,71]. By simultaneously capturing tumor-derived alterations and immune-related signals, ctDNA enables early and sensitive monitoring of response to CAR T-cell therapy while also uncovering resistance mechanisms to guide personalized treatment strategies.

Furthermore, ctDNA is also being explored as a tool for MRD monitoring in DLBCL (Figure 2). Conceptually, MRD detection is key for identifying patients at risk of relapse and for optimizing treatment plans. Serial monitoring of ctDNA after therapy has demonstrated the utility of ctDNA in early relapse detection [35,47]. In a cohort of post-transplant lymphoma patients, including cases of DLBCL, the presence of detectable ctDNA after treatment has been linked to an increased risk of disease progression, even in patients who achieved radiological remission [72]. This suggests that regular ctDNA testing could help identify high-risk patients early, allowing for early re-intervention in case of relapse.

A key enhancement in ctDNA-based MRD detection is the use of phased variant enrichment and detection sequencing (PhasED-Seq). This method increases sensitivity by detecting multiple genetic changes on the same strand of DNA, which helps reduce background noise and improve detection limits [73,74,75]. Seminal results suggest that PhasED-Seq may outperform traditional sequencing methods, such as CAPP-seq, in detecting residual disease. However, further validation is required to establish its superiority and confirm its benefit [35].

Overall, ctDNA monitoring is transforming DLBCL management by providing a minimally invasive and highly sensitive tool for molecular characterization, treatment assessment, and early relapse detection. In clinical trials, ctDNA is increasingly used to monitor treatment response, as its levels can reflect tumor dynamics more rapidly than traditional methods. ctDNA is also becoming integral in identifying MRD, enabling earlier detection of relapses. With ongoing technological progress, ctDNA analysis may be expected to become a part of routine clinical practice in DLBCL, enhancing personalized treatment and improving patient outcomes.

### 2.2. Hodgkin’s Lymphoma

The low prevalence of malignant Hodgkin/Reed–Sternberg (HRS) cells in biopsy samples (ranging from 0.1% to 3%) presents a major challenge for the comprehensive genomic characterization of Hodgkin’s lymphoma (HL) [76]. To address the limitations of tissue-based genotyping, plasma ctDNA has emerged as a reliable source for mutational profiling. In HL patients, the levels of cfDNA are approximately twice as high as those in healthy individuals (~3400 vs. ~1700 hGE/mL of plasma), with a median ctDNA level of around 200 hGE/mL [77]. Despite the tumor cell volume in HL being ten times smaller than that of other aggressive lymphomas, the correlation between ctDNA levels and radiologic tumor volume in HL is strikingly similar to that observed in DLBCL [78]. This suggests that HL may release more ctDNA than DLBCL, likely due to the high rate of apoptosis in HRS cells [79,80].

In a seminal study, ctDNA analysis successfully identified approximately 87.5% of tumor variants found in biopsy samples from 80 newly diagnosed and 32 refractory HL patients, highlighting ctDNA as a potential non-invasive profiling tool [48]. Notably, plasma ctDNA demonstrated a higher median variant allele fraction than biopsy samples, likely reflecting the low tumor cell content characteristic of HL tissue specimens. This finding further emphasizes the value of ctDNA in molecular profiling. Specific mutations detected by ctDNA, such as *XPO1*^E571K^, *STAT6*, and *SOCS*1, can help differentiate HL from other lymphoma subtypes, including DLBCL, primary mediastinal B-cell lymphoma, anaplastic large-cell lymphoma, and mucosa-associated lymphoid tissue lymphoma [53,81,82,83].

Baseline ctDNA levels prior to treatment initiation have been associated with key clinical features, including elevated TMTV, higher Hasenclever international prognostic scores (≥3), increased lactate dehydrogenase (LDH) levels, and advanced disease stage [53,78,84,85]. These correlations suggest that baseline ctDNA could serve as a valuable biomarker integrating conventional prognostic indicators in HL. ctDNA profiling in HL also holds prognostic value. For instance, the detection of the *XPO1*^E571K^ mutation using ddPCR is linked to shorter PFS, with mutation-positive patients showing a 2-year PFS of 57.1%, compared to 90.5% in mutation-negative patients [81]. Similarly, mutations in *TP53* found in ctDNA are associated with poorer PFS (*p* = 0.0038) [86] (Table 1).

Additionally, ctDNA serves as a valuable tool for tracking treatment response and anticipating relapse in HL (Figure 2). Patients with high pretreatment ctDNA levels and persistent ctDNA detection at key time points (C1D15, C3D1, and after four cycles) have significantly shorter PFS [48]. Longitudinal ctDNA monitoring, when combined with PET/CT imaging, detected disease progression in 38% of patients [48]. When both ctDNA and PET/CT results were negative, the negative predictive value was 99%, suggesting that ctDNA could enhance the predictive accuracy of PET/CT in the clinical management of the disease. Furthermore, in advanced HL patients, a 2-log reduction in ctDNA levels after two cycles of ABVD (doxorubicin, bleomycin, vinblastine, dacarbazine) chemotherapy was predictive of CR, supporting a threshold previously validated in DLBCL [48]. Notably, the same prognostic value of ctDNA holds true for immunotherapy-based regimens. In fact, a recent study in previously untreated HL patients receiving pembrolizumab-based therapy demonstrated that ctDNA clearance, observed after the second cycle and at the end of treatment, was strongly associated with improved PFS [87].

Liquid biopsy has also improved the ability to characterize tumor heterogeneity in HL [85]. By integrating somatic copy number alterations with non-silent somatic mutations as weighted features, a dominant genetic clustering approach has revealed two highly stable HL subtypes, namely cluster H1 and H2 [85]. Cluster H1, representing approximately 70% of cases, was defined by somatic mutations in key cell signaling pathways, including NFkB, JAK/STAT, and PI3K. In contrast, cluster H2 (32% of cases) exhibited a broader spectrum of somatic copy number alterations events and mutations in *TP53* and *KMT2D*, genes commonly linked to genomic instability. These molecular distinctions also correlated with clinical features: H2 tumors displayed a characteristic bimodal age distribution and were associated with significantly elevated ctDNA levels, indicative of increased tumor burden and aggressive disease behavior. Importantly, even after adjusting for ctDNA levels, H2 tumors retained negative prognostic implications, reinforcing the independent value of genetic subtyping through liquid biopsy. These findings highlight the growing role of liquid biopsy not only in detecting tumor-derived genomic alterations but also in refining risk stratification and informing therapeutic strategies in HL.

### 2.3. Central Nervous System Lymphomas

DLBCLs involving the central nervous system (CNS), collectively termed CNS lymphomas (CNSLs), are classified into primary and secondary CNS DLBCLs [88]. Primary CNS lymphoma (PCNSL) is a rare and distinct form of NHL, affecting the brain, eyes, leptomeninges, and, in rare instances, the spinal cord [89]. Secondary CNS lymphoma refers to either an isolated relapse of DLBCL within the CNS or concurrent CNS and systemic involvement [90].

Despite significant advancements in radiographic modalities, neuroimaging findings remain suggestive rather than definitive for PCNSL [91]. Consequently, a conclusive diagnosis necessitates stereotactic brain biopsy, incorporating histopathological evaluation and immunohistochemical staining [91]. However, this procedure is both technically challenging and invasive, with a reported failure rate of up to 35% [92]. In this regard, the minimally invasive analysis of ctDNA in CSF or plasma has become a success story in PCNSL, being increasingly integrated into clinical practice as a valuable tool for diagnosis, disease monitoring, and prognostic assessment (Figure 2) [54,55,93,94].

Targeted NGS of plasma-derived ctDNA in patients with PCNSL has shown a sensitivity of 45% in detecting mutations present in the primary tumor tissue, particularly in genes involved in the B-cell receptor signaling pathway, including *MYD88*, *PIM1*, and *CD79* [95,96]. Conversely, ultrasensitive ctDNA profiling detected the presence of ctDNA mutations in 78% of plasma samples and in all CSF samples from CNSL patients prior to treatment [55]. Thus, ctDNA from CSF exhibits a higher concordance with tumor tissue compared to plasma ctDNA, suggesting that CSF may serve as a more reliable medium for genomic analysis in CNS lymphomas [54,55,93]. Two factors limit the sensitivity of detecting specific mutations in plasma or CSF: the concentration of cfDNA and the methods of analysis. Currently, RT-qPCR and NGS are the standard techniques for detecting *MYD88* mutations in PCNSL; however, a direct comparison between these methods has yet to be conducted (Table 1) [5].

The *MYD88*^L265P^ mutation is the most frequently observed genetic alteration in PCNSL biopsies, with a reported prevalence ranging from 52% to 88% [91]. Notably, this mutation is absent in tissue biopsies from brain solid cancers, reinforcing its potential as a distinguishing molecular biomarker for PCNSL [91,97,98]. Several studies comparing PCNSL with DLBCL, inflammatory conditions, and healthy controls have consistently reported a 100% specificity for the detection of *MYD88*^L265P^ in the blood or CSF, highlighting its diagnostic value [99,100,101,102,103,104]. The detection sensitivity of *MYD88^L265P^* is generally higher in CSF compared to plasma. Importantly, Mutter et al. demonstrated high specificity (97%) and sensitivity (100%) for the detection of ctDNA in CSF samples using the PhasED-seq platform, significantly outperforming radiological modalities of response assessment and other less sensitive ctDNA assays [55]. Moreover, Watanabe and colleagues used a ddPCR approach to demonstrate that *MYD88^L265P^* mutations were detectable in diagnostic CSF samples from the overwhelming majority of cases of CNS lymphoma [101]. Moreover, in vitreoretinal lymphomas (VRLs), ctDNA analysis of the *MYD88*^L265P^ mutation in both vitreous fluid and aqueous humor has shown a high concordance rates (69% and 75%, respectively) with cytological findings, supporting its use as a supplementary diagnostic tool for VRL [105,106,107].

Assessing ctDNA pretreatment in PCNSL has also important clinical implications, since undetectable plasma ctDNA at baseline has been linked to favorable outcomes. Notably, in a multivariate analysis, accounting for key clinical and radiographic features known to influence PCNSL prognosis, persistently higher ctDNA levels during treatment were independently and significantly associated with poorer PFS and OS [55]. Furthermore, plasma ctDNA monitoring during treatment allowed for the detection of MRD, identifying patients with a particularly poor prognosis after curative-intent chemoimmunotherapy [55].

Predicting CNS relapse in DLBCL remains a significant challenge beyond PCNSL. Mutations in *MYD88*^L265P^ and *CD79B*^Y196^ have been detected in CSF ctDNA approximately one month before clinical relapse, highlighting their potential for early detection of CNS relapse in lymphoma patients [63,108,109]. However, since 15–20% of CNSL cases do not harbor these mutations, negative results must be interpreted cautiously. A cohort study involving 126 newly diagnosed DLBCL patients and 24 PCNSL patients found that pretreatment CSF ctDNA analysis, using a 475-gene panel related to leukemia and lymphoma, achieved 100% sensitivity and 77.3% specificity in predicting CNS relapse [110]. Moreover, the development of the Molecular Prognostic Index for CNSL (MOP-C) score offers a significant step forward in refining risk prediction. By combining baseline ctDNA detection with MRI-assessed CR at the end of induction therapy, MOP-C builds upon the widely used IELSG score [111]. This enhanced model stratifies patients with CNSL into three distinct relapse risk groups: low risk (0–16.5%), intermediate risk (16.5–50%), and high risk (>50%). Compared to the IELSG score alone, MOP-C demonstrates substantially improved prognostic accuracy, particularly for failure-free survival (FFS) and PFS. Specifically, HR for FFS was 6.60 with MOP-C versus 2.64 with IELSG, and HR for PFS was 3.24 versus 2.37, respectively, thus highlighting the added value of integrating molecular and imaging biomarkers in risk stratification.

### 2.4. Follicular Lymphoma

Follicular lymphoma (FL), the second most common type of NHL, is characterized by significant clinical heterogeneity [112]. While some patients achieve sustained remission with standard therapies, a high-risk subset of patients experience early relapse or transformation to DLBCL, leading to markedly inferior outcomes [112]. Despite significant progress in treatment, accurately stratifying FL patients based on their risk profile remains an ongoing challenge. In particular, identifying individuals with high-risk disease who are prone to early progression or histologic transformation is a critical unmet need. Improved prognostic tools and biomarkers are essential for refining risk assessment, guiding treatment decisions, and optimizing patient outcomes.

Pretreatment ctDNA levels have been proposed as a prognostic indicator in FL, with high levels being independently associated with shorter PFS [56]. cfDNA quantification using ddPCR showed a notable correlation with TMTV, and elevated cfDNA levels were associated with shorter 4-year PFS (73% vs. 94%) [113]. Consistent with these findings, Fernández-Miranda et al. conducted a prospective pilot study using targeted sequencing of plasma-derived ctDNA in 36 patients with FL [114]. The study demonstrated a concordance between mutations detected in ctDNA and those identified in diagnostic tumor biopsies. Moreover, higher pretreatment ctDNA levels were associated with failure to achieve CR and with an increased likelihood of early progression of disease within 24 months (POD24). Interestingly, interim ctDNA levels demonstrated the most significant decrease in patients who achieved CR and in those who did not experience POD24 [114]. These findings collectively underscore the prognostic and dynamic monitoring potential of ctDNA in FL, reinforcing its role in risk stratification and treatment response assessment (Table 1).

The evaluation of MRD in FL has a long-standing history, largely due to the availability of the *BCL2*-IGH rearrangement as a molecular marker, which is present in a very high fraction of FL patients [115]. However, its detection is limited by high rates of somatic hypermutation of IGH genes and by variability in the breakpoint region, and its use for disease monitoring may be affected by the occasional presence of this translocation in healthy individuals. As a result, *BCL2*-IGH rearrangement can only be effectively tracked in approximately 50–60% of FL patients. Liquid-biopsy-based MRD monitoring using ultra-deep sequencing (LiqBio-MRD) shows high sensitivity in detecting trackable somatic mutations in FL. In a study of 84 patients, mutations were identified in 95% of lymph node samples and 80% of baseline liquid biopsies [57]. LiqBio-MRD positivity during first-line therapy correlated with a higher risk of progression, aligning with PET/CT findings. Combined LiqBio-MRD and PET/CT achieved 88% sensitivity and 100% specificity in identifying patients likely to progress within two years [57]. These results highlight the potential of liquid biopsy as a powerful, non-invasive tool for risk stratification and treatment monitoring in FL. ctDNA monitoring has also demonstrated its potential in assessing treatment response across various FL therapies. In patients treated with anti-CD19 CAR T-cell therapy, ctDNA analysis identified MRD-negative status in PET/CT-positive patients, with no relapse after 34 months, suggesting ctDNA as a complementary tool to PET/CT when imaging remains positive for reasons other than lymphoma persistence [57].

Additionally, ctDNA profiling may aid in detecting transformation from indolent FL to aggressive DLBCL, which signifies a poor prognosis [116]. A ctDNA-based prediction model developed by Scherer et al. achieved 83% sensitivity and 89% specificity for identifying histologic transformation, capturing both indolent and aggressive clones before clinical progression [47]. Overall, the results of the currently available literature highlight the value of ctDNA as a minimally invasive biomarker for risk stratification, response assessment, and early detection of disease transformation in FL (Figure 2).

### 2.5. Chronic Lymphocytic Leukemia/Small Lymphocytic Lymphoma

Chronic lymphocytic leukemia (CLL) and small lymphocytic lymphoma (SLL) are characterized by the clonal expansion of mature small CD5+ B-cells and represent different manifestations of the same disease [117]. CLL predominantly affects PB, while SLL mainly involves the lymph nodes. Additionally, monoclonal B-cell lymphocytosis (MBL), especially high-count MBL (B-cell count > 0.5 × 10^9^/L), is now recognized as a precursor condition to CLL/SLL [117,118]. CLL exhibits significant clinical and biological heterogeneity, while SLL, despite its close relationship with CLL, presents unique clinical characteristics [118,119]. The mutational status of immunoglobulin heavy-chain genes, and possibly light-chain immunoglobulin genes as well, helps distinguish various prognostic subsets of CLL and SLL [120,121]. The strides in understanding genomic alterations in CLL/SLL highlight the need for more precise and dynamic molecular monitoring tools. ctDNA has the potential to bridge this gap by offering a non-invasive approach for tracking tumor burden, detecting genetic mutations, and assessing treatment efficacy, which may guide future therapeutic sequencing in CLL [58,122].

The assessment of MRD following treatment in CLL has become increasingly significant, especially with the introduction of time-limited combination therapies [123,124,125]. Undetectable MRD is linked to improved survival outcomes and is considered a key measure for evaluating therapeutic success [126,127]. Although multicolor flow cytometry is commonly used for MRD detection in PB, its limitations in sensitivity for the detection of MRD in other disease compartments underscore the potential relevance of adding alternative approaches [128,129]. The first analysis of ctDNA in CLL has shown promising results for MRD assessment, offering greater accuracy in detecting MRD across multiple compartments [58]. A recent study assessing patient-specific VDJ rearrangements in ctDNA has demonstrated a high concordance in MRD detection between the ctDNA-based method and multicolor flow cytometry. Moreover, incorporating commonly mutated somatic mutations into ctDNA analyses may further refine monitoring of clonal evolution throughout treatment and during disease progression [59,130] (Table 1).

In SLL, where tumor cells are present in both lymph nodes and in the circulation, ctDNA analysis offers a valuable tool for capturing the disease spatial heterogeneity [131,132]. A study on ctDNA in SLL exploited a multiregional sequencing approach across different anatomical compartments, including tissue biopsies, sorted PB CD19+/CD5+ cells, and plasma ctDNA. This study documented that integrating ctDNA analysis may enhance the detection of genetic mutations in individual SLL cases [11]. This approach is particularly relevant for treatment selection, as certain predictive mutations, e.g., *TP53* alterations, may be compartment-specific and thus not detected in the sampled tissue biopsy. These results suggest that combining ctDNA analysis with traditional biopsy and PB assessments can provide a more complete mutational landscape, potentially improving personalized treatment strategies in SLL [11].

## 3. T-Cell Lymphomas

T-cell lymphomas represent a rare and aggressive group of neoplastic disorders characterized by significant heterogeneity and a high degree of invasiveness [133]. Among the various subtypes, extranodal NK/T-cell lymphoma (ENKTL), peripheral T-cell lymphoma, not otherwise specified (PTCL, NOS), nodal T-follicular helper cell lymphoma (nTFHL-AL), and anaplastic large-cell lymphoma (ALCL) account for approximately 80% of all T-cell lymphomas [88].

Conventional tissue biopsy continues to be the gold standard for acquiring molecular data and categorizing lymphoma patients into genetic subtypes [133]. However, its applicability is often limited in cases where surgical intervention is impractical, such as deeply localized disease, critically ill patients, unresectable tumors, or individuals with low procedural compliance, or when the content of neoplastic cells is low. These facts highlight a need for minimally invasive diagnostic approaches, particularly for prognostic assessment, treatment monitoring, and evaluation of drug resistance in T-cell lymphoma (Figure 2). In this context, liquid biopsy is emerging as a promising alternative, offering a non-invasive method for genetic profiling and disease surveillance, thereby expanding clinical opportunities for personalized patient management, with most efforts focusing on ENKTL and PTCL [134].

### 3.1. ENKTL

ENKTL exhibits higher circulating cfDNA concentrations and lower mutant allele frequency compared to DLBCL [135]. ctDNA effectively detects tumor-specific genetic alterations, with a high concordance between mutations identified in ctDNA and those found in tumor biopsy samples, highlighting its role as a non-invasive tool for diagnosis and precision medicine. In more detail, in high-risk ENKTL, targeted sequencing of 31 mutated genes identified *STAT3* mutations as the most frequent alteration (29%), followed by *TP53* (21%), *BCOR*, *KMT2D* (17%), and *NOTCH1* (12%) [136]. Additionally, pretreatment ctDNA levels have been shown to correlate with Ann Arbor stage and serum LDH levels, underscoring the potential as a biomarker for tumor burden [137].

cfDNA and ctDNA analyses have emerged as promising prognostic tools in ENKTL, providing valuable insights into disease progression, treatment response, and recurrence. In high-risk ENKTL patients, targeted NGS of tumor tissue and longitudinal plasma ctDNA showed that lower cfDNA concentrations were associated with improved survival outcomes, with a significantly higher one-year PFS rate (90.0% vs. 36.4%) [136]. Furthermore, patients who exhibited rapid clearance of ctDNA mutations achieved significantly higher CR rates (80.0% vs. 0%) and demonstrated more favorable PFS (79.0% vs. 20.0%) compared to those with persistently detectable mutations [136]. Additionally, specific genetic mutations (i.e., *KMT2D* and *DDX3X*) detected in ctDNA correlate with poor prognosis, further reinforcing its utility in tumor prognostication and disease monitoring [60]. These findings emphasize the role of cfDNA/ctDNA analysis in defining MRD status, offering a non-invasive approach for prognostic assessment and guiding therapeutic decision-making in ENKTL (Table 1).

### 3.2. PTCL

In PTCL, ctDNA analysis exhibits a high concordance with tumor tissue biopsies in detecting genetic alterations. Consistently, in one PTCL series, the overall sensitivity of plasma cfDNA for detecting biopsy-confirmed mutations was 73.9%, with a specificity of 99.6% [61]. Specifically, the detection of mutations in key genes such as *TET2*, *RHOA*, *DNMT3A*, and *IDH2* in plasma ctDNA has enabled a minimally invasive approach for diagnosing nTFHL-AT, a distinct PTCL subtype. This advancement highlights the potential of ctDNA in facilitating early diagnosis and molecular characterization, ultimately improving disease management and personalized treatment strategies. Additionally, ctDNA sequencing has uncovered novel mutations in the *RHOA* gene, including c.73A>G (p.Phe25Leu) and c.48A>T (p.Cys16)*, which were subsequently validated via Sanger sequencing. This advancement underscores the potential of ctDNA in early disease detection, molecular characterization, and personalized treatment strategies.

Beyond its diagnostic utility, ctDNA has shown promise in disease monitoring and prognostication. Analysis of plasma ctDNA mutation profiles in 94 patients with PTCL using targeted NGS revealed a significant association between post-treatment ctDNA levels and survival outcomes [138]. Additionally, assessment of serial serum samples using NGS-based T-cell receptor sequencing during treatment revealed that 38% of patients achieved ctDNA clearance after two cycles of therapy, while 46% still had detectable ctDNA at the end of treatment [139]. Persistent ctDNA detection post-therapy was associated with a trend toward poorer survival outcomes. Similarly, a longitudinal study of PTCL demonstrated a strong correlation between post-treatment ctDNA mutation burden and disease recurrence or progression [139]. Notably, a genomic equivalent (GE) reduction exceeding 1.5 log from baseline to treatment completion was significantly associated with improved survival compared to patients with a GE reduction of less than 1.5 log [138]. Collectively, these findings suggest that ctDNA holds promise as a non-invasive tool for monitoring treatment response and predicting clinical outcomes in PTCL patients, with growing evidence supporting its utility, particularly in high-risk subgroups.

## 4. ctDNA in Lymphoma Clinical Trials

Several completed clinical trials have explored the role of ctDNA in the clinical management of lymphomas. In an early-phase trial involving patients with R/R DLBCL (NCT03311958), ctDNA was employed as a biomarker to detect early relapse following chemotherapy [66]. Patients with detectable ctDNA received nivolumab maintenance therapy for two years to avoid complete relapses, highlighting the potential of ctDNA as an early marker to guide therapeutic decisions. Additionally, the observational study NCT06777290 aimed to compare the diagnostic performance of ctDNA with standard imaging techniques, including whole-body MRI and PET/CT, for staging and prognostication in DLBCL, supporting ctDNA’s utility as a complementary or alternative diagnostic tool [140]. Another observational study (NCT05066555) linked to the FIL-Rouge trial evaluated ctDNA genotyping for monitoring treatment response in advanced-stage HL patients [141]. This study focused on the concordance between ctDNA analysis and interim PET/CT in assessing CR, underscoring the potential of ctDNA to enhance response evaluation. In the context of PCNSL, the NCT06454266 trial monitored dynamic changes in ctDNA in CSF before and after treatment with a novel combination regimen (orelabrutinib, rituximab, and methotrexate). The results of this trial provide insights into the real-time treatment response in this rare and aggressive lymphoma subtype [142]. Collectively, these studies support the growing role of liquid biopsy as a transformative tool in lymphoma management, enabling more precise, real-time, and non-invasive monitoring. Ongoing clinical trials (Table 2) are further evaluating the clinical value of ctDNA, particularly for tracking disease progression and guiding therapeutic decisions.

## 5. Discussion and Conclusions

Liquid biopsy techniques based on ctDNA have rapidly advanced from a promising research tool to a clinically relevant biomarker with multifaceted potential applications across lymphoid malignancies. Its non-invasive nature, coupled with high sensitivity and specificity, enables real-time insights into the assessment of tumor burden, molecular heterogeneity, and treatment dynamics that are often missed by conventional imaging or tissue biopsies. Across studies, ctDNA has demonstrated significant prognostic value, since baseline levels often correlate with disease burden and outcomes, and early clearance during or after treatment predicts favorable survival. This temporal resolution allows for dynamic response assessment, making ctDNA a powerful tool for MRD monitoring, with the potential to detect molecular relapse months before clinical or radiographic evidence emerges. Despite these promising advances, several significant challenges must be overcome before ctDNA can be fully adopted in routine clinical practice. Among these are the lack of standardized and validated methods for ctDNA detection, as well as issues of limited sensitivity and potential false positives, particularly due to confounding factors such as clonal hematopoiesis of indeterminate potential (CHIP). Moreover, diagnostic thresholds, prognostic cut-offs, and optimal sampling frequency remain unresolved, hindering the consistent interpretation of results across studies and patient populations. Nonetheless, substantial strides have been made in improving ctDNA technologies. Consistently, advancements in sequencing technologies, such as CAPP-seq and PhasED-Seq, have further improved the sensitivity and specificity of ctDNA analysis, allowing for the detection of patient-specific mutations and clonal evolution, the assessment of tumor spatial heterogeneity, and the identification of treatment-resistant subpopulations without the need for repeated invasive procedures. Importantly, integrating ctDNA analysis with standard imaging modalities (i.e., PET/CT or MRI) offers a promising, minimally invasive strategy to enhance risk stratification and guide treatment. This combined approach could help identify patients at high risk of early relapse who may benefit from more innovative therapeutic options and/or consolidation and maintenance therapy. In parallel, ctDNA has shown promise in guiding novel therapies, such as CAR T-cell treatment, by enabling early response prediction and patient stratification based on molecular risk profiles. In the era of precision medicine, the trajectory of ctDNA research points toward its expanding role as a central pillar in lymphoma management, shaping the future of risk-adapted therapy, clinical trial design, and truly personalized oncology.

## Figures and Tables

**Figure 1 ijms-26-04869-f001:**
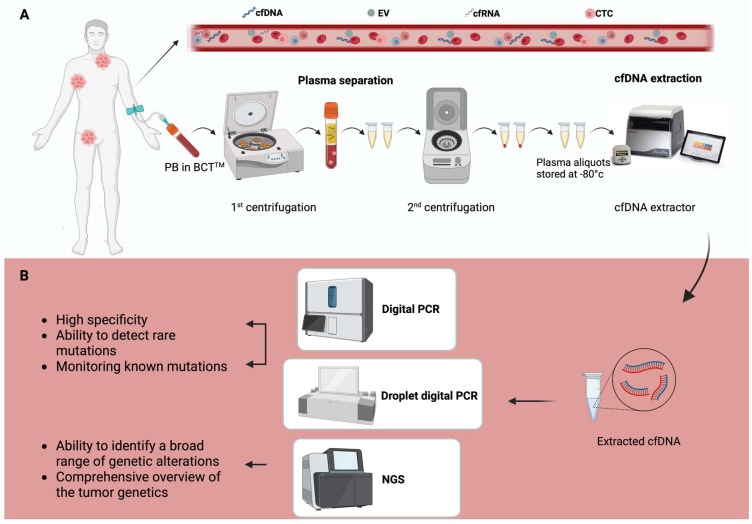
Techniques for cfDNA detection in lymphomas. (**A**) Overview of cell-free DNA (cfDNA) isolation from peripheral blood (PB). The patient’s PB is collected in blood collection tubes (BCT^TM^), followed by a two-step centrifugation process to separate plasma. The plasma is then aliquoted and stored at −80 °C until cfDNA extraction is performed using an automated extractor. (**B**) Summary of the primary detection methods for cfDNA. Digital PCR and droplet digital PCR provide high specificity for detecting known mutations, while next-generation sequencing (NGS) offers comprehensive genetic profiling and detailed tumor characterization. Created in BioRender.com. Gaidano, G. (2025).

**Figure 2 ijms-26-04869-f002:**
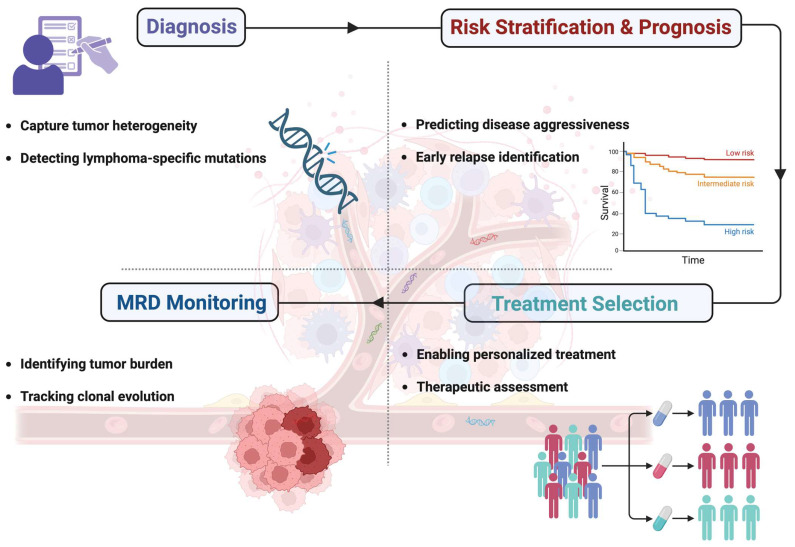
The multifaceted role of liquid biopsy in lymphoma: from diagnosis to relapse detection. cfDNA analysis plays a critical role in multiple aspects of lymphoma management. At diagnosis, liquid biopsy detects tumor heterogeneity and lymphoma-specific mutations. Concerning risk stratification and prognosis, liquid biopsy allows for the assessment of disease aggressiveness and the risk of relapse. At the time of treatment selection, liquid biopsy enables personalized therapy. Finally, liquid biopsy provides a tool for minimal residual disease (MRD) monitoring, thus tracking disease clonal evolution. Created in BioRender.com. Gaidano, G. (2025).

**Table 1 ijms-26-04869-t001:** Clinical applications of ctDNA in lymphomas.

Lymphoma Subtype	References	Origin of ctDNA	Patient Count	Technologies	Sensitivity	Clinical Application
DLBCL	Rossi et al., 2017 [6]	Plasma	30	CAPP-Seq	~10^−3^ ctDNA detected in 66.6%	Disease genotyping
	Kurtz et al., 2018 [52]	Plasma	217	CAPP-Seq	Detected in 98%	Early relapse prediction
	Moia et al., 2025 [46]	Plasma	166	CAPP-Seq	95.8% concordance with tissue biopsy	Molecular clustering
HL	Spina et al., 2018 [48]	Plasma	112	CAPP-Seq	~10^−3^ Detected in 81.2%	Monitoring of clonal evolution and treatment response assessment
	Camus et al., 2021 [53]	Plasma	94	dPCR, NGS	-	MRD monitoring and risk stratification
CNSL	Bobillo et al., 2021 [54]	CSF	19	Targeted NGS, ddPCR	-	Disease genotyping
	Mutter et al., 2023 [55]	CSF, plasma	92	CAPPSeq, PhasED-seq	Detectable in 78% of plasma and 100% of CSF	Treatment response assessment and risk stratification
FL	Sarkozy et al., 2017 [56]	Plasma	34	NGS-based immunosequencing	Detected in 86%	Prognostic biomarker
	Jiménez-Ubieto et al., 2023 [57]	Plasma	84	LiqBio-MRD	Detected in 80%	Treatment monitoring
CLL	Yeh et al., 2019 [58]	Plasma	32	dPCR, TS	Detected in 80%	MRD assessment
	Fürstenau et al., 2022 [59]	Plasma	46	dPCR	-	MRD assessment
T-cell lymphomas	Li et al., 2020 [60]	Plasma	65	Targeted sequencing	93.8% concordance with tissue biopsy	Guiding therapeutic decision-making
	Wei et al., 2023 [61]	Plasma	47	NGS	73.9% concordance with tissue biopsy	Mutation profiling and disease monitoring

*DLBCL*, diffuse large-B-cell lymphoma; *ctDNA*, circulating tumor DNA; *CAPP-seq*, cancer personalized profiling by deep sequencing; *HL*, Hodgkin’s lymphoma; *dPCR*, digital PCR; *NGS*, next-generation sequencing; *MRD*, minimal residual disease; *CNSL*, central nervous system lymphoma; *CSF*, cerebrospinal fluid; *ddPCR*, droplet digital PCR; *PhasED-seq*, phased variant enrichment and detection sequencing; *FL*, follicular lymphoma; *LiqBio-MRD*, liqui- biopsy-based MRD monitoring using ultra-deep sequencing; *CLL*, chronic lymphocytic leukemia; *TS*, targeted amplicon deep sequencing.

**Table 2 ijms-26-04869-t002:** Ongoing clinical trials evaluating the role of ctDNA in lymphomas.

Disease	Methods	Endpoints	Study Design	Identifier
**PCNSL**	ctDNA	Evaluate ctDNA conversion rate in CSF	Interventional (Phase II)	NCT04401774
**DLBCL**	ctDNA and PET	Evaluate a PET/CT- and ctDNA-oriented therapy in DLBCL to test treatment response	Interventional (Phase II)	NCT04604067
**MCL**	Flow cytometry, PCR and ctDNA	Estimate MRD negative response rates by aggregate measure of peripheral blood/or bone marrow flow cytometry, PCR, and ctDNA	Interventional (Phase II)	NCT04234061
**PCNSL**	ctDNA	Characterizing ctDNA for early response assessment	Observational	NCT06755619
**DLBCL**	ctDNA	Measure ctDNA in real time during standard treatment	Interventional	NCT06693830
**HL**	ctDNA	MRD assessment by ctDNA	Observational	NCT05254821
**PTCL**	ctDNA	Evaluate the feasibility of ctDNA measurement in blood plasma for treatment evaluation and MRD surveillance	Observational	NCT06362148
**DLBCL, MCL, and FL**	ctDNA	To determine the clinical utility of ctDNA as well as CTC-based MRD assessment in CAR therapy patients	Observational	NCT05255354
**LBCL**	ctDNA	Early assessment of lymphoma treatment response using ctDNA analyzed by phased variant analysis	Observational	NCT06550427

*PCNSL*, primary central nervous system lymphoma; *ctDNA*, circulating tumor DNA; *CSF*, cerebrospinal fluid; *DLBCL*, diffuse large-B-cell lymphoma; *PET/CT*, positron emission tomography/computed tomography; *MCL*, mantle cell lymphoma; *MRD*, minimal residual disease; *HL*, Hodgkin’s lymphoma; *PTCL*, peripheral T-cell lymphoma; *FL*, follicular lymphoma; *CTC*, circulating tumor cells; *LBCL*, large-B-cell lymphoma.

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
