# Peer review of "Liquid Biopsy in B and T Cell Lymphomas: From Bench to Bedside"

_ijms, 2025, doi:10.3390/ijms26104869_

Round 1

Reviewer 1 Report

Comments and Suggestions for Authors

This manuscript provides a timely overview of ctDNA-based liquid biopsy as an emerging tool for lymphoma diagnosis, molecular profiling, and disease monitoring. The authors convincingly highlight ctDNA’s advantages over tissue biopsy—particularly in capturing tumor heterogeneity, enabling sequential monitoring, and overcoming sampling limitations in Hodgkin lymphoma (HL) and primary CNS lymphoma (PCNSL). While the clinical potential is evident, broader standardization and validation are needed before routine implementation. This is an interesting manuscript, but I have several following concerns:

  1. Add the table comparing ctDNA utility across lymphoma subtypes (sensitivity, key mutations, clinical applications).
  2. Figure illustrating ctDNA’s role in the lymphoma care continuum (diagnosis → therapy → follow-up).

  3. Abbreviations should be defined when they first appear in the text. Such as "MYD88, CD79B, EZH2" in Line 18, "LDH" in Line 286, ...please double check all the text and revsie them.
  4. Figures directly quoted or modified from other published works need to apply for copyright or obtain a support code for publication.
  5. Avoid crossing pages in the tables presented in the text as much as possible.
  6. Please unify the format of references in the article, including the author's name, the case of words in the title of the article, the writing of the name of the journal, and the page number.

Author Response

Reviewer1: This manuscript provides a timely overview of ctDNA-based liquid biopsy as an emerging tool for lymphoma diagnosis, molecular profiling, and disease monitoring. The authors convincingly highlight ctDNA’s advantages over tissue biopsy—particularly in capturing tumor heterogeneity, enabling sequential monitoring, and overcoming sampling limitations in Hodgkin lymphoma (HL) and primary CNS lymphoma (PCNSL). While the clinical potential is evident, broader standardization and validation are needed before routine implementation. This is an interesting manuscript, but I have several following concerns:

1- Add the table comparing ctDNA utility across lymphoma subtypes (sensitivity, key mutations, clinical applications).

2- Figure illustrating ctDNA’s role in the lymphoma care continuum (diagnosis → therapy → follow-up).

3- Abbreviations should be defined when they first appear in the text. Such as "MYD88, CD79B, EZH2" in Line 18, "LDH" in Line 286, ...please double check all the text and revise them.

4- Figures directly quoted or modified from other published works need to apply for copyright or obtain a support code for publication.

5- Avoid crossing pages in the tables presented in the text as much as possible.

6- Please unify the format of references in the article, including the author's name, the case of words in the title of the article, the writing of the name of the journal, and the page number.

We thank the reviewer for her/his constructive comments.

1- A new table titled “Clinical Applications of ctDNA in Lymphomas” has been added to the manuscript to summarize and clarify the key clinical uses of ctDNA in this context (see page 13).

2- Figure 2 previously illustrated the role of ctDNA in lymphoma management. However, it has been revised to adopt a more continuous and descriptive format, providing a cohesive overview of liquid biopsy applications in the clinical management of lymphomas.

3- To enhance readability and avoid lengthy phrases that might distract from the core concepts, we have included in the supplementary materials a comprehensive list of all gene abbreviations used in the manuscript. LDH has been spelled out in the text (see line 292).

4- Both Figure 1 and Figure 2 are original and were created using licensed BioRender.com. We have included the appropriate publication license with our submission. Additionally, a statement acknowledging that the figures were created with BioRender has been added to the respective figure legends.

5- Crossing pages in the tables has been avoided

6- We have thoroughly reviewed the manuscript, and the reference format has been unified in accordance with the journal's guidelines throughout the article. This includes consistency in authors' names, title capitalization, journal names, and page numbers.

Reviewer 2 Report

Comments and Suggestions for Authors

This manuscript presents a comprehensive and timely review of the current and emerging applications of circulating tumor DNA (ctDNA) in the management of B-cell and T-cell lymphomas. The title accurately reflects the scope of the review, and the abstract effectively summarizes the key points, emphasizing the value of liquid biopsy as a complementary tool to tissue biopsy.

The introduction offers a well-structured background on the clinical heterogeneity of lymphomas and the limitations of traditional tissue-based diagnostics. The authors correctly highlight the need for minimally invasive approaches that can capture tumor heterogeneity and enable longitudinal monitoring, thus justifying the focus on ctDNA. The discussion of the biological properties of ctDNA and the technical considerations for its extraction and analysis (e.g., plasma vs. serum, double-spin centrifugation, digital PCR, NGS) is well-detailed and demonstrates a solid understanding of the field. However, the section would benefit from a clearer comparison of sensitivity, specificity, and limitations of the different analytical techniques, as well as a brief discussion of cost and accessibility barriers to broader clinical implementation.

The core of the review—application of ctDNA in specific lymphoma subtypes—is its strongest section. The authors provide a thorough and up-to-date overview of ctDNA’s clinical utility across DLBCL, Hodgkin lymphoma, CNS lymphomas, follicular lymphoma, CLL/SLL, and T-cell lymphomas. The content is well-organized and substantiated with numerous references to landmark studies. The discussion of ctDNA’s prognostic and predictive value in DLBCL, including its integration with LymphGen classification, PhasED-seq, and CAR-T cell therapy response prediction, is particularly commendable. The review also appropriately highlights how ctDNA enables minimal residual disease (MRD) assessment and the detection of early relapse.

In the section on Hodgkin lymphoma, the authors effectively address the challenge of low tumor cell content in biopsy specimens and demonstrate how ctDNA can serve as a reliable surrogate for molecular profiling. The correlation between ctDNA levels and PET/CT outcomes is well-described, though additional emphasis on the limitations of current evidence (e.g., small sample sizes or lack of prospective validation) would improve balance. Similar comments apply to the sections on PCNSL and T-cell lymphomas, which provide strong summaries of current evidence, but could benefit from more critical discussion of diagnostic specificity and practical barriers to widespread clinical adoption.

The section on ongoing and completed clinical trials is highly informative and adds translational value to the review. Table 1 effectively summarizes the current landscape of clinical research focused on ctDNA applications in lymphomas. A few clarifying details—such as which trials are interventional versus observational, or which endpoints are surrogate markers—would further enhance its utility.

The discussion and conclusions are concise yet forward-looking. The authors correctly identify ctDNA as a powerful biomarker for personalized lymphoma care and argue for its integration with standard imaging tools. However, a more explicit discussion of existing challenges—such as lack of assay standardization, regulatory hurdles, and limited reimbursement—would provide a more critical and complete view of the field.

In terms of presentation, the manuscript is generally well-written, though a few minor language refinements (for example, avoiding phrases like “seminal study” without further context) would improve clarity. The scientific style is consistent and appropriate, and the manuscript is supported by extensive and current references.

Although the present review focuses on lymphoid malignancies, the inclusion of comparative insights from solid tumors could provide useful translational context. In particular, the article (Cancers, 2021; DOI: 10.3390/cancers13133373) offers a comprehensive overview of liquid biopsy applications in prostate cancer, including analytical challenges, assay standardization, and the clinical significance of ctDNA for monitoring treatment resistance and disease progression. Referencing this article would help broaden the discussion by illustrating how ctDNA is being operationalized in another tumor type, which may inform strategies for its clinical integration in lymphomas. Drawing such parallels could support the manuscript's call for wider adoption and standardization of ctDNA in hematologic settings.

In summary, this is a valuable and well-executed review that provides an in-depth, clinically relevant, and up-to-date synthesis of ctDNA applications in lymphoma. With minor revisions to enhance critical analysis and incorporate comparative perspectives, the manuscript would make a strong contribution to the literature.

Author Response

Reviewer 2: This manuscript presents a comprehensive and timely review of the current and emerging applications of circulating tumor DNA (ctDNA) in the management of B-cell and T-cell lymphomas. The title accurately reflects the scope of the review, and the abstract effectively summarizes the key points, emphasizing the value of liquid biopsy as a complementary tool to tissue biopsy. 

The introduction offers a well-structured background on the clinical heterogeneity of lymphomas and the limitations of traditional tissue-based diagnostics. The authors correctly highlight the need for minimally invasive approaches that can capture tumor heterogeneity and enable longitudinal monitoring, thus justifying the focus on ctDNA. The discussion of the biological properties of ctDNA and the technical considerations for its extraction and analysis (e.g., plasma vs. serum, double-spin centrifugation, digital PCR, NGS) is well-detailed and demonstrates a solid understanding of the field. However, the section would benefit from a clearer comparison of sensitivity, specificity, and limitations of the different analytical techniques, as well as a brief discussion of cost and accessibility barriers to broader clinical implementation.

The core of the review—application of ctDNA in specific lymphoma subtypes—is its strongest section. The authors provide a thorough and up-to-date overview of ctDNA’s clinical utility across DLBCL, Hodgkin lymphoma, CNS lymphomas, follicular lymphoma, CLL/SLL, and T-cell lymphomas. The content is well-organized and substantiated with numerous references to landmark studies. The discussion of ctDNA’s prognostic and predictive value in DLBCL, including its integration with LymphGen classification, PhasED-seq, and CAR-T cell therapy response prediction, is particularly commendable. The review also appropriately highlights how ctDNA enables minimal residual disease (MRD) assessment and the detection of early relapse. 

In the section on Hodgkin lymphoma, the authors effectively address the challenge of low tumor cell content in biopsy specimens and demonstrate how ctDNA can serve as a reliable surrogate for molecular profiling. The correlation between ctDNA levels and PET/CT outcomes is well-described, though additional emphasis on the limitations of current evidence (e.g., small sample sizes or lack of prospective validation) would improve balance. Similar comments apply to the sections on PCNSL and T-cell lymphomas, which provide strong summaries of current evidence, but could benefit from more critical discussion of diagnostic specificity and practical barriers to widespread clinical adoption. 

The section on ongoing and completed clinical trials is highly informative and adds translational value to the review. Table 1 effectively summarizes the current landscape of clinical research focused on ctDNA applications in lymphomas. A few clarifying details—such as which trials are interventional versus observational, or which endpoints are surrogate markers—would further enhance its utility.

The discussion and conclusions are concise yet forward-looking. The authors correctly identify ctDNA as a powerful biomarker for personalized lymphoma care and argue for its integration with standard imaging tools. However, a more explicit discussion of existing challenges—such as lack of assay standardization, regulatory hurdles, and limited reimbursement—would provide a more critical and complete view of the field.

In terms of presentation, the manuscript is generally well-written, though a few minor language refinements (for example, avoiding phrases like “seminal study” without further context) would improve clarity. The scientific style is consistent and appropriate, and the manuscript is supported by extensive and current references.

Although the present review focuses on lymphoid malignancies, the inclusion of comparative insights from solid tumors could provide useful translational context. In particular, the article (Cancers, 2021; DOI: 10.3390/cancers13133373) offers a comprehensive overview of liquid biopsy applications in prostate cancer, including analytical challenges, assay standardization, and the clinical significance of ctDNA for monitoring treatment resistance and disease progression. Referencing this article would help broaden the discussion by illustrating how ctDNA is being operationalized in another tumor type, which may inform strategies for its clinical integration in lymphomas. Drawing such parallels could support the manuscript's call for wider adoption and standardization of ctDNA in hematologic settings. 

In summary, this is a valuable and well-executed review that provides an in-depth, clinically relevant, and up-to-date synthesis of ctDNA applications in lymphoma. With minor revisions to enhance critical analysis and incorporate comparative perspectives, the manuscript would make a strong contribution to the literature. In this narrative review the authors present current literature on the use of liquid biopsies in B and T cell lymphomas. The review covers a broad overview of the process of liquid biopsy, its benefits and what tumor derived components can be isolated. Figure 1 is clear and provides an easy to understand explanation of the process.

We thank the reviewer for her/his comments.

1- A clearer emphasis on the limitations of ctDNA has been incorporated into the Discussion section (see lines 630–637) to provide a more balanced and comprehensive perspective.

2- A summary table of ongoing clinical trials involving ctDNA in lymphomas has already been included in the manuscript (see table 2, page 16). The table details the type of each trial (observational vs. interventional) and outlines their respective endpoints, offering a comprehensive overview of the current clinical research landscape in this area.

3- To provide translational context, we have added a concise overview in the Introduction section outlining the role of liquid biopsy in solid tumors (see lines 48–54). Additionally, the referenced work (Cancers, 2021; DOI: 10.3390/cancers13133373) has been cited to further enrich the manuscript.

Reviewer 3 Report

Comments and Suggestions for Authors

In this narrative review the authors present current literature on the use of liquid biopsies in B and T cell lymphomas. The review covers a broad overview of the process of liquid biopsy, its benefits and what tumor derived components can be isolated. Figure 1 is clear and provides an easy to understand explanation of the process.

The majority of the review focuses on the use of cell-free RNA (cfRNA). There is a good explanation of the general biological basis of ctDNA as well as a disease specific description for different types of lymphoma. The authors describe the uses of ctDNA for both diagnostic and  monitoring purposes and this is well described in figure 2.  There is an appropriate reference to specific data from research papers within the main text.

The authors have included a good mix of preclinical and clinical data throughout. There are sufficient recent citations included in the text and no obvious omissions of relevant citations. The writing is of a very high quality and the style is easy to digest.

Overall this is an excellent review article.

Author Response

Reviewer3: The majority of the review focuses on the use of cell-free RNA (cfRNA). There is a good explanation of the general biological basis of ctDNA as well as a disease specific description for different types of lymphoma. The authors describe the uses of ctDNA for both diagnostic and  monitoring purposes and this is well described in figure. There is an appropriate reference to specific data from research papers within the main text. The authors have included a good mix of preclinical and clinical data throughout. There are sufficient recent citations included in the text and no obvious omissions of relevant citations. The writing is of a very high quality and the style is easy to digest.

Overall this is an excellent review article.

We thank the reviewer for her/his encouraging and positive comments.

Round 2

Reviewer 1 Report

Comments and Suggestions for Authors

The authors have addressed all my concerns, I recommend accepting it in current form.